# Environmental Viability Analysis of Connected European Inland–Marine Waterways and Their Services in View of Climate Change

**Sándor A. Némethy** [1,2,3,*], **Anna Ternell** [2], **Lennart Bornmalm** [4], **Bosse Lagerqvist** [1] and **László Szemethy** [5]

1 Department of Conservation, University of Gothenburg, 40530 Gothenburg, Sweden; bosse.lagerqvist@conservation.gu.se
2 Institute of Geography and Earth Sciences, Faculty of Natural Sciences, University of Pécs, 7624 Pécs, Hungary; anna.ternell@pe.se
3 Department of Natural Sciences, Lórántffy Institute, University of Tokaj, 3950 Sárospatak, Hungary
4 Department of Marine Sciences, University of Gothenburg, 40530 Gothenburg, Sweden; lennart.bornmalm@marine.gu.se
5 Institute of Biology, Faculty of Natural Sciences, University of Pécs, 7624 Pécs, Hungary; szemethy.laszlo@pte.hu
* Correspondence: sandor@conservation.gu.se

**Abstract:** Inland waterways and their connections to marine transport systems constitute a substantial resource for the establishment of green infrastructures, flood prevention, and environmental conservation. However, these developments have numerous inherent environmental hazards such as water and air pollution, a loss of habitats, increased coastal erosion, the transfer of invasive species between connected watercourses and lakes, and the transport of pollutants through watercourses to coastal areas. Climate change may aggravate these environmental problems through changing temperatures, reduced precipitation, enhancing the adverse impact of excess nutrient discharge, and the entry of invasive species. In this study, we analyse the main European inland waterway corridors and their branches to assess the ecological viability of a pan-European inland waterway network. The environmental viability of such network depends on the right assessment of ecosystem services and protection of biodiversity. A model structure for landscape conservation, green infrastructure development, water replenishment, and ecosystem reconstruction is proposed, considering a sustainable combination of multimodal inland waterway and rail transport.

**Keywords:** inland waterways; ecosystem services; water level fluctuations; flooding; invasive species; greenhouse gas emissions; alternative fuels; habitat reconstruction

## 1. Introduction

Inland waterway transport is one of the most important land transport systems, together with road and rail transport. The mostly fossil fuel-based road transport has vast adverse impacts on the environment, such as greenhouse gas emissions and other air pollutants enhancing climate change, noise, health risks, and infrastructure, which has serious impacts on the landscape and the natural ecosystems [1]. Therefore, increasing the share of the inland waterway and rail transport might be a far more sustainable solution [2]. In this study we intend to raise issues and propose solutions for using and developing inland waterways from a more complex, ecosystem-centred viewpoint by seeking answers to the following questions:

1.  Which European surface waters are and can be considered in the future as inland waterways, what are their functions, and how do inland waterways connect different aquatic ecosystems?
2.  Why are inland waterways vulnerable and how can their ecological resiliency be preserved or increased?
3.  To what extent may an increase of inland waterway transport reduce greenhouse gas and particulate matter emissions compared to rail transport?
4.  What are the ecological consequences of new connections between different watercourses created to eliminate the "missing links" between waterway corridors?
5.  What are the possibilities to replace fossil fuels with electricity in inland waterway transport compared with railway?
6.  What are the most important differences and similarities between European inland waterways and the waterway systems in North America in terms of environmental, economic, and social benefits and problems?
7.  Is there an ecologically sustainable solution for a pan-European inland waterway network?

In 2006, the European Conference of Ministers of Transport [3] noted that in spite of large increases in transport demands, nearly all of these volumes were absorbed by land-bound systems, and most were significantly road-based. This was due to the failure of inland waterways to attract new traffic flows. In 2016, the Innovation and Networks Executive Agency of the European Commission [4] stated that this imbalance between road transport and inland waterways was still predominant, and the prediction for 2030 was that the road freight transport was projected to increase by around 40%. This constitutes a severe problem since the EU transport policy has as its main aim the development of more energy-efficient systems. Therefore, one of the key goals the European Commission has formulated for 2050 is that a 50% shift of intercity cargo and passenger transport from road to rail and waterborne transport should take place. The concept of a pan-European inland waterway network has been created (Figure 1).

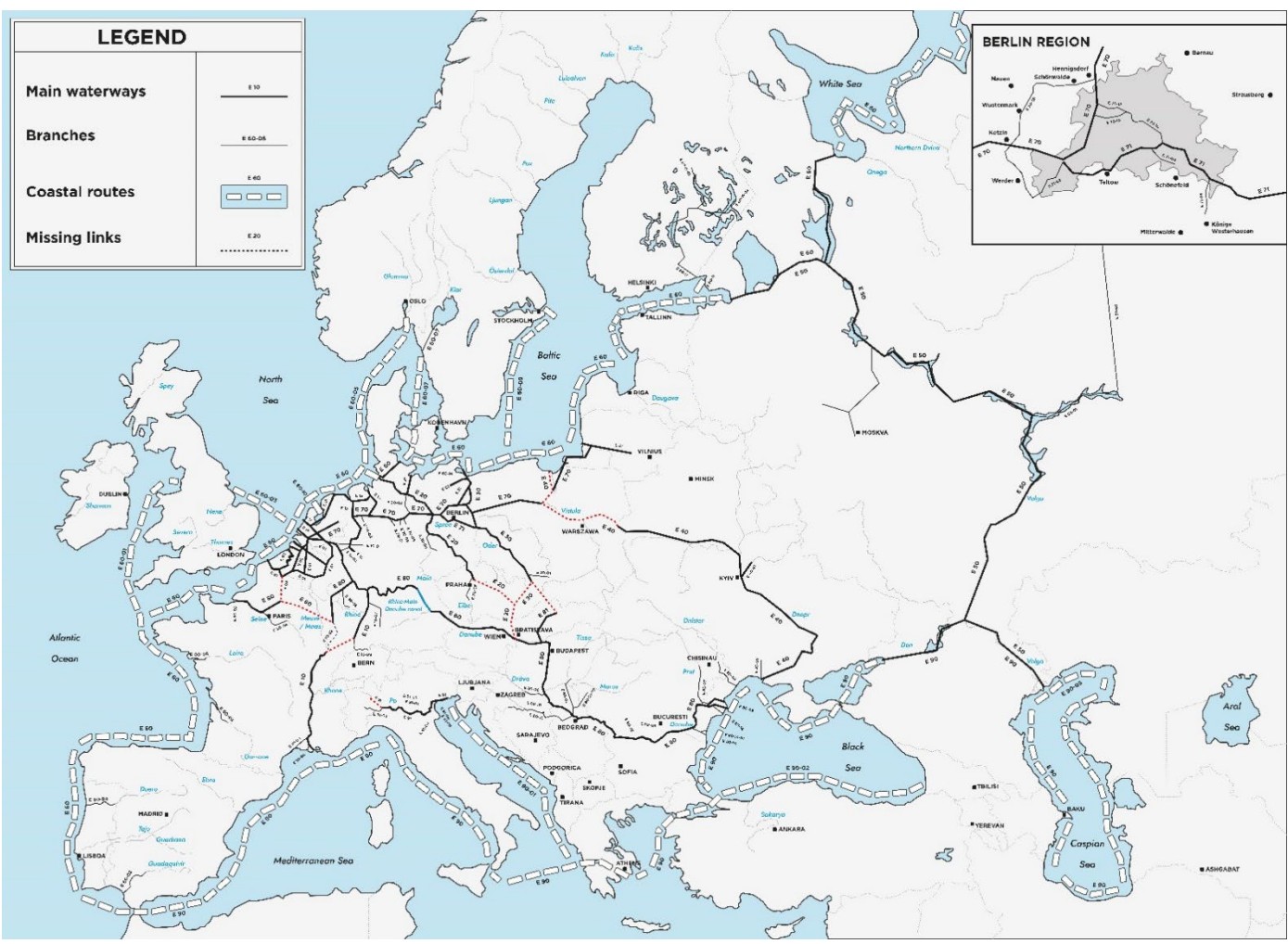

**Figure 1.** Map of the potential pan-European inland waterway network showing how the E40 waterway between the Baltic and the Black Sea could function—the only problem is the absence of crucial connections between watercourses: the missing links are marked with dotted lines on the map. Furthermore, this system can integrate the water transport networks of Belarus and Ukraine with the EU (Source: redrawn and modified after https://unece.org/ accessed on 8 January 2022).

## 2. Materials and Methods

Although this study is mainly a review, we are presenting here a new, integrated concept of inland waterways, taking into consideration contradicting opinions regarding the environmental impact of rail and waterway transport. Materials used for the study include reports from the European Commission, from the Innovation & Networks Executive Agency (INEA) of the European Union, relevant OECD reports, proceedings of the European Conference of Ministers of Transport, a substantial number of Internet sources, scientific articles, book chapters, and results from our earlier research concerning the Lake Balaton–Sió–Danube connection, which can be regarded today as a potential inland waterway. The main focus of this study is the Rhine–Main–Danube waterway and its possible extensions, such as the Danube–Oder–Elbe Canal including the Váh River, the technical possibilities of these extensions, and even the possible development of the Lake Balaton–Sió–Danube waterway, as an example of the multifunctional use (e.g., tourism, agriculture, water level control) of inland waterways (Figure 2).

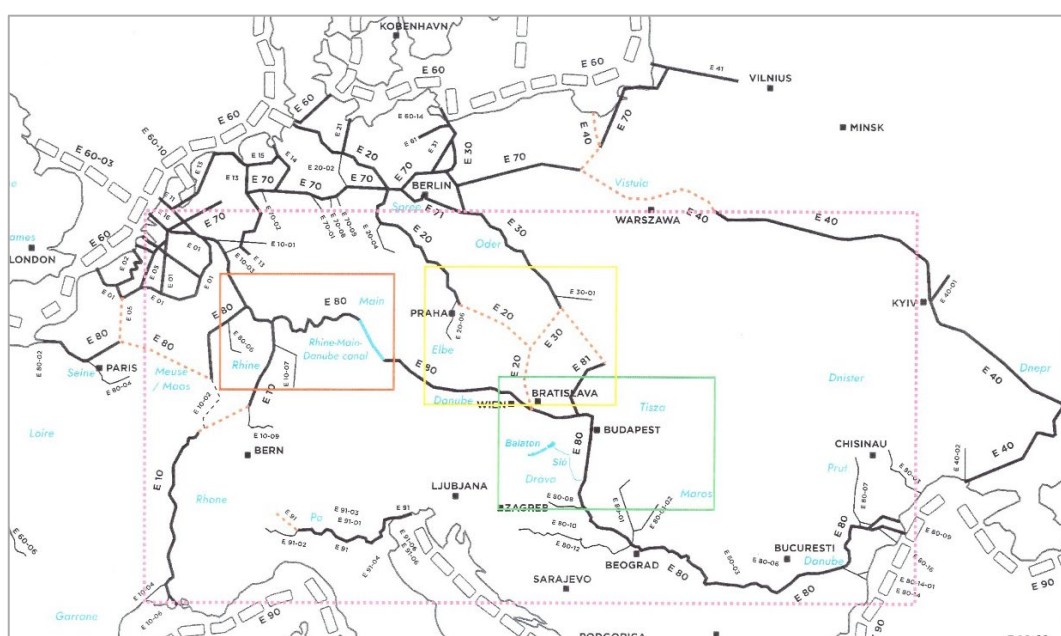

**Figure 2.** Master figure showing the locations of key areas mentioned in this study. Legend: [ ] The Rhine–Main–Danube Canal; [.......] The hydrographic catchment areas of the Rhine, Rhône, Po, and Danube rivers draining the Alps; [ ] The concept of the Danube–Oder–Elbe Canal, including the Váh River, resulting in a substantial NE extension of the Rhein-Main-Danube corridor; [ ] Map of Hungary with Lake Balaton and the Sió canal linking the lake to the river Danube. Source: redrawn from Figure 1.

The key issues of our analysis include the ecological impact assessment of riverbed regulations and alterations of flow regimes, the connection of different habitats through the construction of canals to eliminate the "missing links", and the ecological and economic impact of climate change in terms of water supply, water level fluctuations, and the conservation of biodiversity on both the ecosystem and landscape levels.

Even if our study is dealing with European inland waterway systems, it is important to make some short, relevant comparisons with other systems in other continents and climate zones, such as the vast inland waterway systems of North America. There are other waterway systems, such as the Amazon River system in South America, the Nile and Zambezi rivers in Africa, the Yangtze River in China, the Ganges in India, the Mekong in Vietnam, and the Volga in Russia, which are of interest for further studies, but the scope of this study allows to restrict our comparisons to one continent with well-developed waterways: in this case, North America, mainly the USA. This is particularly important in view of climate change, increasing environmental pressure, and increasing population in the vicinity of watercourses and lakes. The other reason for our choice is that there has been an increasing number of extreme weather events reported from Europe and the USA (e.g., tornadoes, wildfires, floods, droughts, etc.—many of them occur much more frequently in the USA), which can be attributed both to climate change and the increasing capacity of developed countries to observe, measure, and report these events [5]. Our method is based on a holistic, ecosystem-centred approach, which serves as the theoretical basis for the proposed planning concept of inland waterway networks where all ecological, social, and economic factors are taken into consideration.

## 3. Which European Surface Waters Are and Can Be Considered in the Future as Inland Waterways, What Are Their Functions, and How Do Inland Waterways Connect Different Aquatic Ecosystems?

Inland waterways include canals, rivers, and lakes linked into larger systems, connecting terrestrial areas and marine coastal regions, which connect inland waterways. From the viewpoint of ecosystems and ecosystem services, inland waterways are highly

multifunctional, as they represent more than surface watercourses and lakes, providing freight transport between inland settlements and marine coastal areas. As all surface—and even subsurface—waters, both natural and artificial, inland waterways are valuable, ecologically sensitive aquatic habitats, water supplies for agriculture, industry, and drinking water, and landscape-forming factors, and many of them are popular tourist destinations. Inland waterways connect several aquatic and terrestrial, natural, and artificial ecosystems such as lakes, artificial fishing lakes and ponds, smaller surface watercourses in their catchment area, wetlands, forests, grasslands, agro-ecosystems, estuaries and brackish water ecosystems, and marine coastal areas (Figure 3).

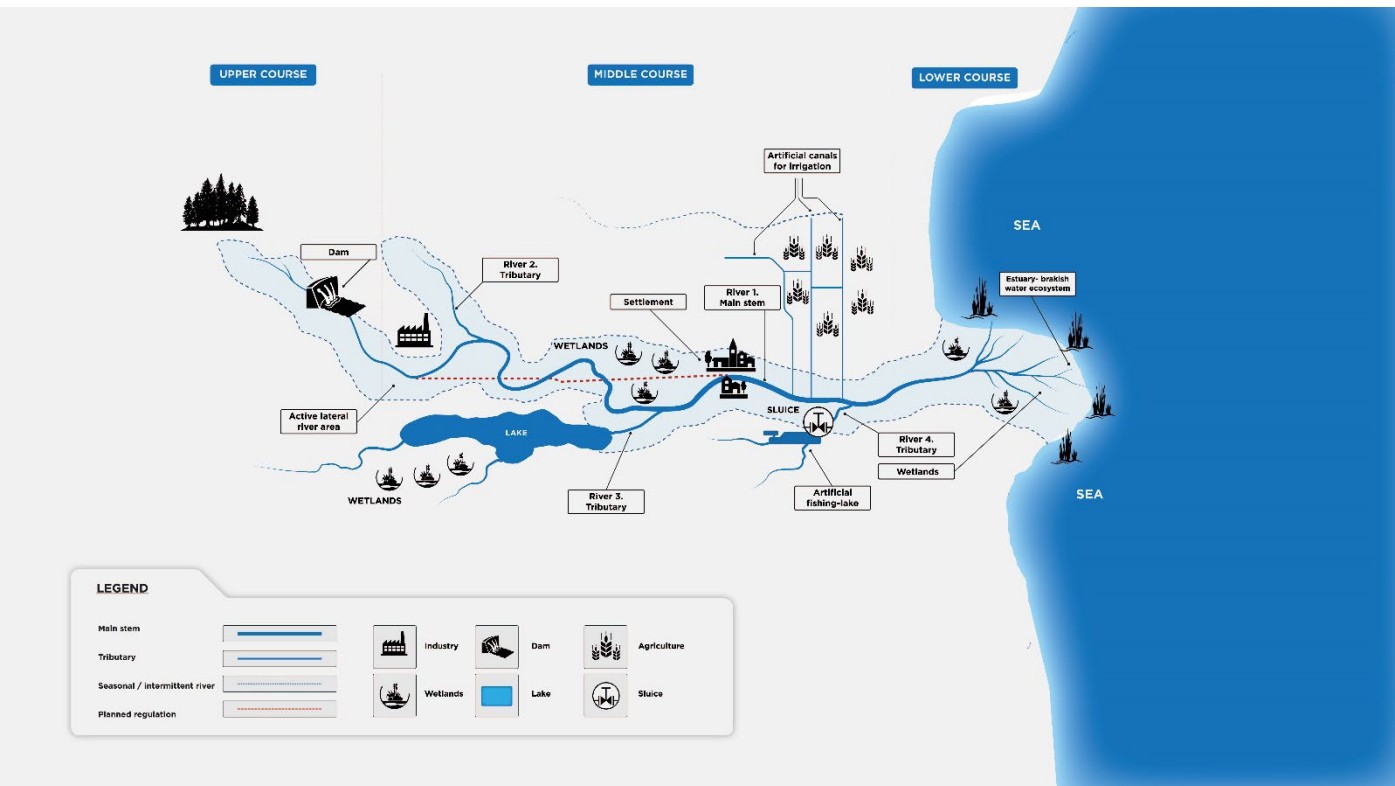

**Figure 3.** The structure of larger rivers, which can be used as inland waterways connecting natural and artificial ecosystems. It is important to consider how possible regulation and changes in land use might affect the natural ecosystems connected by these rivers. Source: own design.

The European inland waterway network of 41,000 km connects 25 Member States. Europe has set ambitious environmental and climate policy goals, from carbon neutrality to a circular economy to cleaner air to cleaner transport, such as using and further developing inland waterways and reducing the fossil fuel-based highway transport. This solution has potential benefits in terms of cost savings, reduced pollution, and an environmentally feasible increase in economic growth. The new EU strategies will have to eliminate the infrastructure bottlenecks in order to improve the efficiency of European inland waterway transport. The geographical distribution of Europe's population is advantageous regarding the vicinity to inland waterways or coastal areas since more than half of the population lives within or near these regions where most European industrial centres are accessible through inland waterways. The main international inland waterways in Europe are (Figure 1):

1.  The Rhine–Danube network (Rhine/Meuse–Main–Danube inland waterway axis);
    a.  Rhine Basin
    b.  Danube Basin
2.  The central European canal and river network (including the Weser, Elbe, and Oder rivers);

3. The Azo–Black–Caspian Seas basin;
4. The Czech–Slovak network;
5. Coastal routes and connected inland waterways.

Since the opening of the Main–Danube Canal in 1992, the transcontinental Rhine–Main–Danube Waterway (Figures 1, 2, and 4) has allowed vessels to travel 3500 kilometres from Rotterdam on the North Sea to the port of Sulina on the Black Sea, becoming one of the largest waterways in the world (This means a navigable inland waterway system consisting of several rivers, not the longest river. The longest river in the world is the Nile, with a total length of 6650 km from East Africa to the Mediterranean). It has a length of 3483 km and, together with its associated river sections, a waterway system of 12,000 km.

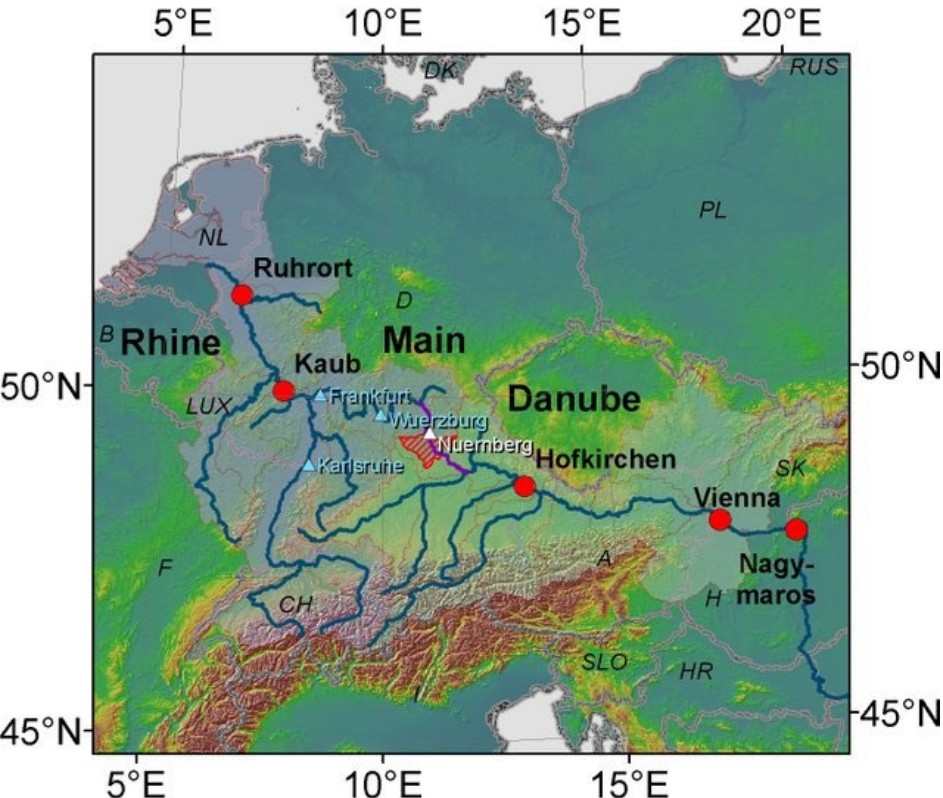

**Figure 4.** The Rhine–Main–Danube transport corridor, showing the Main–Danube Canal. Legend: ━━━ the Main–Danube canal [6].

## 4. Why Are Inland Waterways Vulnerable and How Can Their Ecological Resiliency Be Preserved or Increased?

The modification of inland waters to create waterways for transportation may result in several, often detrimental, consequences for aquatic habitats. However, by applying suitable green infrastructures and wetland reconstruction, these adverse impacts may be sufficiently counterbalanced. The vulnerability of inland waterways is a complex issue including both natural and anthropogenic impacts. One of the most important factors is climate change, which has several adverse impacts, such as water level changes, eutrophication and low water quality, reduced biodiversity, expansion of invasive species, freshwater shortage (both for household and industry)—altogether creating a substantial loss of vital ecosystem services [7]. For transportation, the most important problem is the reduction of water levels caused by climate change-induced droughts, which can make navigation impossible or force operators to substantially reduce the load of the vessels. In addition to drought, extreme flooding can impair the quality and capacity of waterways. Extreme weather events, such as rapidly increasing precipitation or the fast melting of glaciers, often cause floods during which water levels exceed the maximum permitted

ones [8,9]. Furthermore, during hard winters, the formation of ice, particularly on slow-flowing rivers can seriously disturb or totally inhibit inland navigation, which happened on the river Danube in 2005 and 2006 [10].

*4.1. Structural Problems in European Waterways*

The lack of adequate infrastructure (bottlenecks, missing links) is the main obstacle to inland waterway transport. The most common types of missing connections include:

*Bridges.* The height of the free space under bridges and the distance between the pillars determine the size of inland waterway vessels and how many layers of containers they can carry. The height of the free gauge decreases at high water levels and increases at low water levels.

*The morphology of the waterways.* It depends on the width and shape of the fairway and whether and at what speed upstream and downstream ships can pass at the same time. The depth of the draft available on a waterway determines how many tonnes of goods can be carried on an inland waterway vessel. The size of the draft loaded has a decisive effect on the cost-effectiveness of inland waterway transport.

*Locks.* The lock capacity can extend the voyage time, as the waiting time depends on the size of the ship or the ship's convoy passing through the locks. Single-chamber locks can cripple all river traffic if even one is closed for maintenance.

*Missing links.* Parts of inland waterways are future networks of international importance that do not yet exist. An example is the lack of an important connection between the *Seine* in France and the *Schelde* in Belgium. The Member States are concerned, and TEN-T is already addressing the issue.

Bottlenecks need to be eliminated in order to improve the navigability of rivers and thus remove the main infrastructural obstacles to the development of European inland waterway transport. An interesting case is the German section of the Danube between *Straubing* and *Vilshofen,* which is crucial for the entire inland waterway network. As previous studies analysing alternative options for removing bottlenecks were discussed by various stakeholders, including environmentalists, the European Commission and the German authorities decided to commission a new, detailed, neutral analysis study to analyse the potential costs and environmental impacts of the two most likely options:

Option A, which had a slightly lower environmental impact and promised a less comprehensive improvement in navigability conditions, and

Option C 2.80, which targeted better navigability conditions for slightly more significant environmental impacts. The ecological compensation area was 1360 ha for variant "A" and 1415 ha for variant "C 2.80". The study examined the issue so thoroughly that detailed technical plans were prepared for both variants. Although option "C 2.80" results in better navigability conditions and a better cost-benefit ratio, the German authorities have decided to implement option "A", which does not provide the navigability conditions for the sustainable development of European inland waterway transport (https://www.eca.europa.eu/Lists/ECADocuments/SR15_01/SR15_01_HU.pdf, accessed on 25 January 2022). In this case, the direct environmental benefits gained higher priority, but it is very likely that from a long-term perspective, the environmental benefits would have outweighed the short-term drawbacks in terms of reduced emissions, less use of roads, and reduced costs.

The complex ELOHA method can be helpful in the construction of ecologically viable improvements to inland waterways. The name of the Nature Conservancy ELOHA (Ecological Limits of Hydrologic Alteration) method suggests that it seeks to explore the ecological effects of hydrological variability [11]. ELOHA consists of the following steps:

1. hydrological foundation (daily water flow data measured for each section of the river for the original and current conditions, e.g., after flooding, water use data);

2.  river (section) typing (based on ecologically important watercourse parameters, water quality characteristics, riverbed shape and material, and species composition of associations [12]);
3.  establishing links between water transport and ecological status for different river types (selection of ecological indicators through the processing of expert opinions, verification of the application of ecological principles by case studies);
4.  preparation, adoption, and implementation of the necessary water management policy decisions (consultations, determination of the order of priority of ecosystem functions and ecological risks, recording of permitted hydrological changes).

*4.2. Ecological Aspects of the Rhine—Danube Corridor*

Since the opening of the Main–Danube Canal in 1992, the transcontinental Rhine–Main–Danube Waterway allows vessels to travel 3500 kilometres from Rotterdam on the North Sea to the port of Sulina on the Black Sea, connecting a vast area of inland locations due to their partially navigable tributaries. The fact that this connection is not fully functional is due to the substantial differences between the riverbed morphologies of the Rhine and the Danube: the riverbed of the Rhine is relatively narrow but deep, while the riverbed profile of the Danube is characterized by wide and shallow conditions. As a result, deeper draft vessels with a higher carrying capacity have become widespread in Western Europe, but they are only able to navigate the Danube for a part of the year (depending on water levels) with depth restrictions. In order to ensure the unobstructed and fast movement of "Rhine-type" deep-sea vessels with a large transport capacity on the Danube (particularly on the Hungarian Danube section) at least 343 days a year, the natural physical conditions of the river would have to be significantly modified. Thus, the Rhine Basin has the highest population density and accounts for more than 80% of all inland waterway freight transport, while approx. 9% of the total inland waterway transport takes place on the Danube and the Rhine–Main–Danube canal [6, 13].

The Hungarian section and the floodplain of the Danube are also famous on a Central European level for their valuable wildlife and rare habitats, which are very vulnerable. Among the habitats rich in natural values, the willow, poplar, alder, and ash populations, which still occur in their natural states along the Danube, are of great importance at the European level. Furthermore, the Danube and its floodplain affect several nature conservation areas:

Seven protected natural areas of national significance (2 national parks, 2 landscape protection areas, 3 nature protection areas), covering the area of operation of 4 national park directorates);
12 Natura 2000 sites (8 priority nature conservation and 4 bird protection areas);
4 forest reserves;
2 Ramsar sites;
2 biosphere reserves.

The Danube riverbed is entirely part of the national ecological network area and has different classifications: a core area, an ecological corridor, and a buffer area [14]. To provide the parameters needed for the development of navigation, the most significant burden would be the damming of the river, causing a general change in the ecological system of the river that is unacceptable from a nature conservation point of view. The existence of the still partially natural habitats and functioning ecological processes is due to the fact that the Hungarian Danube section has remained a free-flowing river without dams. Living communities associated with flowing water do not tolerate slowing down via flooding, and the changed conditions would jeopardize the survival of many habitats and natural assets.

*4.3. Impact of Climate Change*

Although the global mean temperature has increased only by 0.8 °C compared with pre-industrial times, Europe had to face greater warming than the rest of the world, which resulted in more flooding, heavy rain events, and major droughts in recent decades. The most catastrophic droughts occurred in the summer of 2003 in central parts of Europe and in 2005 in the Iberian Peninsula. Furthermore, the melting of the European glaciers has shown substantial acceleration since 1980 [15], and rivers draining the alps (Figure 5) receive an extra amount of water from these glaciers. Although there are several uncertainties concerning climate projections, there are certain opinions that global warming could be even beneficial for inland navigation in some regions. The average economic benefit from the decrease in low water levels could reach € 8 million annually by the end of the century [16], based on ice-free periods and a reduction of floods in middle and southern Europe from an annual precipitation reduction of up to 20%, while in northern Europe, precipitation increasing by 10% to 40% causes increasing trends of floods. Thus, the impact of global warming on inland waterways cannot be considered as beneficial in the long term, taking into account the fact that the temporary benefits of the reduction of low water levels in one place may coincide with floods and damage to waterways and ecosystems somewhere else.

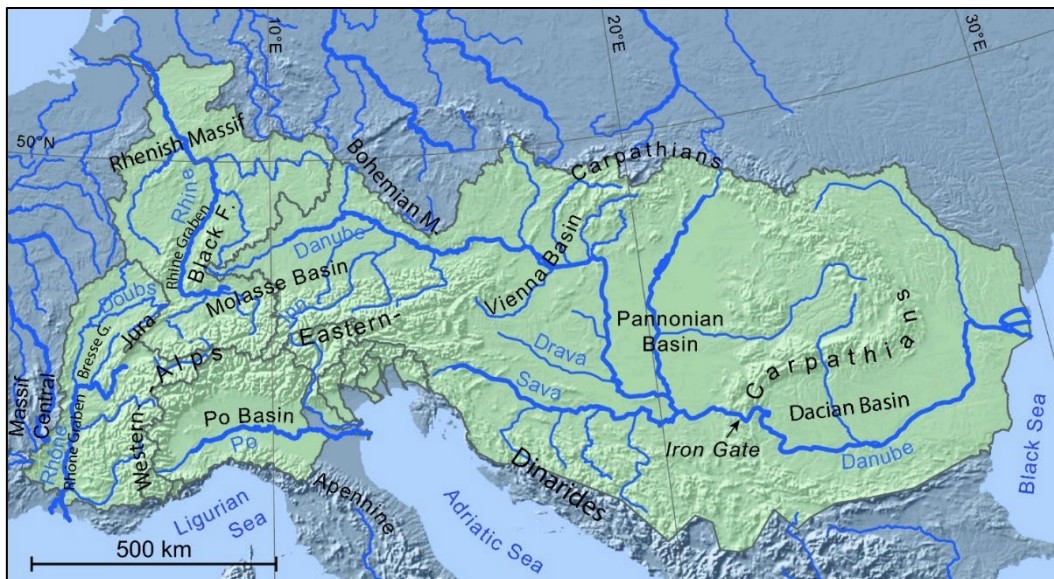

**Figure 5.** The hydrographic catchment areas of the Rhine, Rhône, Po and Danube rivers draining the Alps [17].

When estimating the impact of climate change on inland waterways, short-term economic factors may often bias the proposals due to a one-sided income-loss approach not sufficiently dealing with the environmental consequences of infrastructure development and changes of land use or the planning of water replenishment measures and other human interventions to ensure suitable water depth and safe navigation. Regarding the changing morphology of riverbeds due to fluctuating water levels, river-bank erosion, sediment transport, and variable inflow from tributaries on certain segments of the waterways, discharges should be considered location-specific. The other problem is substantial uncertainty in terms of predictability of fluctuations in temperature, precipitation, river discharge, and loss of ecosystem services. Therefore, a more complex approach is required to use discharges for relevant types of analyses instead of water levels [6].

## 5. To What Extent May an Increase of Inland Waterway Transport Reduce Green-House Gas and Particulate Matter Emissions Compared to Rail Transport?

Shipping can usually be economical if the quantity of goods corresponding to the maximum capacity of the vessel is available and if it has to be transported over long distances. The quantity of goods offered for transport depends on the market, but typically, bulk and heavy cargoes enter the transport market in large quantities (by weight). Due to globalization, transport distances may also increase, so inland waterway transport can be a competitive alternative. While the economical minimum distance for shipping used to be around 800–1000 km a few decades ago, today, with the development of loading technology, this can be much smaller at only a few hundred km. This is especially true for geographical areas with extensive waterway systems (e.g., Belgium, The Netherlands, Germany). In addition, inland navigation is excellent for transporting oversized, overweight, individual cargo. One of the reasons for the economy is the low specific energy consumption of inland waterway vessels compared to other transport vehicles. This is due to the favourable useful weight/total weight ratio and the relatively low travel resistance due to the low speed.

Regarding the environmental impact of transport vessels on inland waterways, the average energy consumption of upstream and downstream courses should be considered when the clear long-term advantage of inland navigation over other modes appears to be obvious, although some studies argue in favour of railway transport [18]. However, considering the source of electricity in railway transport, the energy source (fossil, renewable, nuclear) used by the operating power plants must be taken into account in order to get a more realistic base of comparison. The environmental friendliness of inland navigation in terms of air pollution is partly due to the low specific energy consumption, as the number of harmful substances emitted into the air is directly proportional to fuel consumption [19], which greatly depends on the applied propulsion technology. Air pollutants are basically divided into two main groups: greenhouse gases that enhance climate change ($CO_2$, $CH_4$, $N_2O$, $SF_6$, and other fluorinated gases) and other air pollutants (mainly nitrogen oxides—NOx, sulphur oxides—SOx, and particulate matter—PM). To compare the actual and specific energy consumption and the emissions of $CO_2$, NOx, PM, and $SO_2$ among different transport modes, researchers analysed a specific transportation task: the transportation of 300 TEU (TEU = 20-foot equivalent unit, a quite inexact (but generally accepted) unit to measure cargo capacity) on the Budapest–Constanta route [19]. One alternative (A1) was road transport, with modern EURO3 environmental grade semi-trailers. The second version (A2) was inland water transport, a modern self-propelled vessel and a pushed barge with a total capacity of 2 × 150 TEU. The third option examined (A3) was also an inland waterway solution, with an old pusher boat with heavy-duty engines, a lower speed, and two pushed barges with a total capacity of 300 TEU. The fourth option (A4) was electric rail transport of 25 wagons/train with 3 TEU/wagon capacity. The length of the road and rail route was approx. 1000 km, while that of the inland waterway was 1400 km. The relatively high emission for electric rail transport comes from the use of fossil fuels for producing electric power. Thus, taking into consideration the energy consumption, there is a clear long-term advantage of inland navigation over other modes if fossil fuels are used (Figure 6).

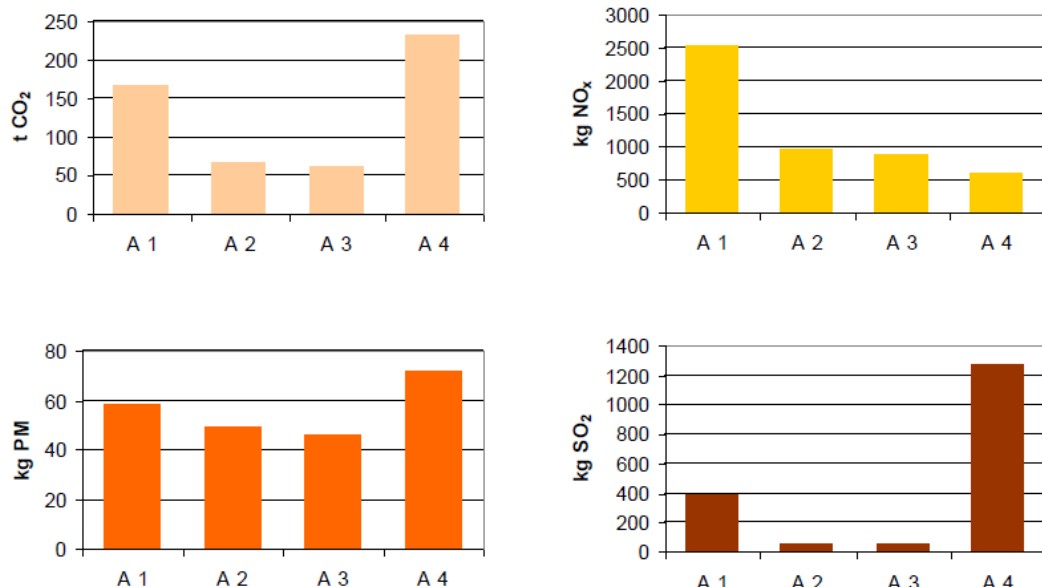

**Figure 6.** Modelling the differences in air pollution between four different transport alternatives (A1 road transport, A2 and A3 inland waterways, A4 railway), inland waterways (options A2 and A3) appear to be the most environment friendly. The high level of emissions for rail transport is due to fossil fuel-based power plants for electric traction [19].

However, some other authors [20] prefer rail transport to waterways since if the electricity production was mainly renewable and nuclear, the emission for rail transport would be minimal, only a fraction of the here estimated volume and, in fact, better than the emission data of inland waterway transport. According to more recent studies (https://www.eea.europa.eu/publications/rail-and-waterborne-transport, accessed on the 28 January 2022), in terms of emissions, $CO_2$ emissions from inland waterway transport can be estimated at 30–40 g $CO_2$/tkm (tkm = tonne kilometre); the average specific emissions of $CO_2$ from rail transport are 15–20 $CO_2$/tkm for electric traction and 35–40 g $CO_2$/tkm for diesel traction. Thus, the average specific emissions for rail transport with electric traction are substantially less than the emissions of inland waterway transport (Figure 7) (https://cedelft.eu/publications/methodology-for-ghg-efficiency-of-transport-modes/, accessed on 30 January 2022).

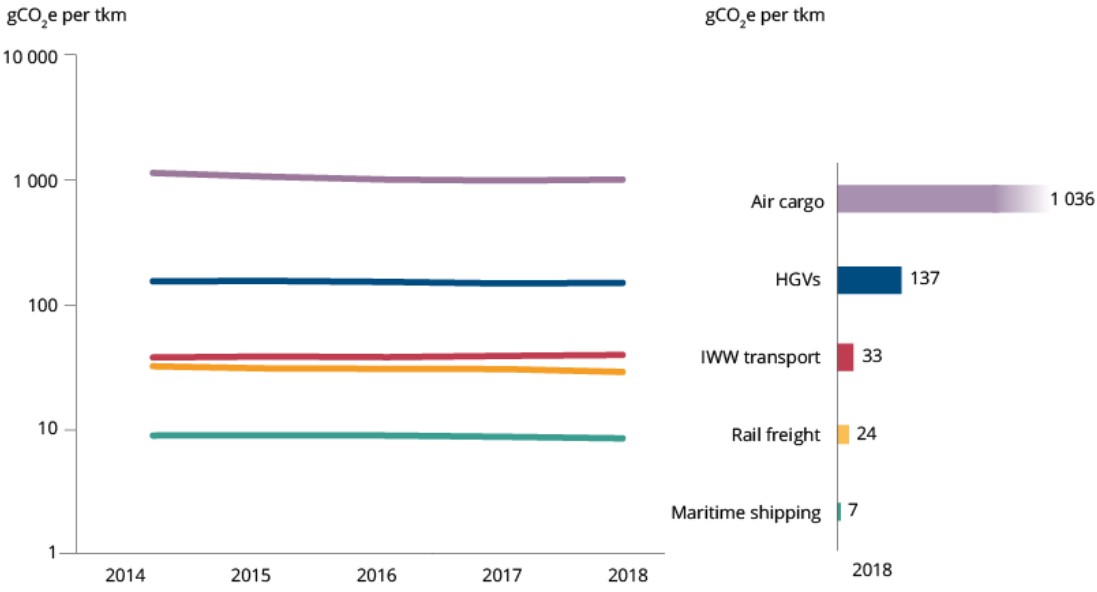

**Figure 7.** Average GHG emissions by motorised mode of freight transport, EU-27, 2014–2018. Source: Fraunhofer ISI and CE Delft, 2020 https://www.eea.europa.eu/publications/rail-and-water-borne-transport accessed on 30 January 2022).

It is also worth mentioning that on inland waterways, most often, bulk goods are transported by ship (raw materials: coal, lignite, petroleum; agricultural, hunting, and forestry products), which is also typical of the transport of goods by rail. Overall, inland waterway transport consumes twice as much energy as electrified rail transport and emits approximately twice as much $CO_2$. In addition, increased ship traffic is associated with increased pollution (e.g., oily bottom water discharges, port pollution), and in the event of an accident, the pollution of living waters causes irreversible damage.

Concerning water pollution, there are strict requirements for both seagoing vessels navigating on inland waterways and river-sea vessels, according to the International Convention for the Prevention of Pollution from Ships (MARPOL 73/78) (https://www.imo.org/en/About/Conventions/Pages/International-Convention-for-the-Prevention-of-Pollution-from-Ships-(MARPOL).aspx, accessed on 2 February 2022). Furthermore, the river basin authority may introduce more stringent pollution control for inland waterways when this is justified in special cases, such as drinking water supply or environments of high ecological sensitivity. However, regarding the protection of water quality, not only the transport vessels but also the ports and the adjacent service networks, such as shipyards for maintenance and reparation and pollution from adjacent human settlements, shall be taken into consideration.

Another important impact on the environment is traffic noise; however, noise emissions from inland waterway transport are relatively low compared to other modes of transportation, and traffic is far from populated areas. Therefore, inland waterways have a great environmental advantage due to the lack of noise pollution compared with other ways of transport.

Although the European Commission's strategy outlines the integration of inland waterways into the intermodal transport chain together with rail and short-sea-shipping due to their environmental advantages (https://eur-lex.europa.eu/legal-content/EN/TXT/?uri=celex%3A52011DC0144, accessed on 15 April 2022), the inland waterway infrastructure can be developed only in an ecologically sustainable way, since the ecologically sensitive river and adjacent lacustrine and marshland ecosystems must be preserved or, in some cases, restored. Therefore, development projects shall operate within the framework of European environmental laws, including the Birds and Habitats Directives and the Water Framework Directive (WFD), which emphasize the creation of a Natura 2000 network to protect habitats and often endangered species. This requires an integrated multidisciplinary approach and an all-inclusive stakeholder management based on community participation.

**6. What Are the Ecological Consequences of New Connections between Different Watercourses Created to Eliminate the "Missing Links" between Waterway Corridors?**

It is important to point out that river regulation also involves water pollution, significantly changed hydrologic conditions (e.g., slowed down by damming, damage to the natural filter layer) significantly reduce the self-cleaning capacity of the river. As a result, drinking water supplies may be threatened, and the ecosystem services provided by the river may be significantly reduced. Furthermore, connecting different waterways can result in an uncontrolled transfer of invasive species not only to other watercourses but also to lakes and marshland ecosystems. Without disputing the need to improve navigability and the minimization of interventions, only development ideas adapted to the characteristics of the river can be supported to protect their environmental and natural values and ecological status.

*6.1. Controversial Plans: Building a New Waterway to Eliminate a Missing Link: The Danube–Oder–Elbe Canal–Including the Váh River*

From the point of view of water management, transport, energy, and tourism, the Czech Republic would benefit from the construction of the Danube–Oder–Elbe Canal (https://saveoder.org/en/eeb-meta-news-controversial-plans-for-destructive-danube-oder-elbe-waterway-moving-forward/, accessed on 13 February 2022), according to a feasibility study commissioned by the Czech Ministry of Transport on a concept that has been much debated for decades. The new waterway would start north from the section of the Danube between Bratislava and Vienna and connect to the Oder on Polish territory over Ostrava in northern Moravia. There would be a westbound branch about 40–50 km below Ostrava that would connect the canal to the Elbe (Figure 8). According to the study, the length of the new waterway, which will be built on rivers in the regions, would be about 2075 kilometres. The cost of connecting the three rivers was estimated at 16 billion Euros. It is envisaged that some of the work could be funded by the European Union. According to the Czechs, the construction of the canal would employ at least 60–70 thousand people for several years and would significantly help transport and freight throughout the Central European region. However, when examining the environmental consequences of this project, we can find many reasons against it. We ask the following: Is this an investment to strengthen the economies of the Czech and Central European states, or to build another megalomaniac construction that causes unnecessary environmental damage?

What are the potential benefits of a Danube–Oder–Elbe Canal?

a.  Economic recovery in the Czech Republic and other countries concerned through this large-scale investment (job creation, etc.);
b.  The connection of the Czech Republic to waterways of European importance so that a larger area of the country could have access to the sea;
c.  Moravian–Silesian industrial companies (ironworks) would be better off delivering their products to other parts of the world;
d.  Some of the freight transport by road (20–30%) would shift to cheaper and less environmentally damaging water transport (at least in terms of air pollution);
e.  Possibility of more effective flood protection in some sections.

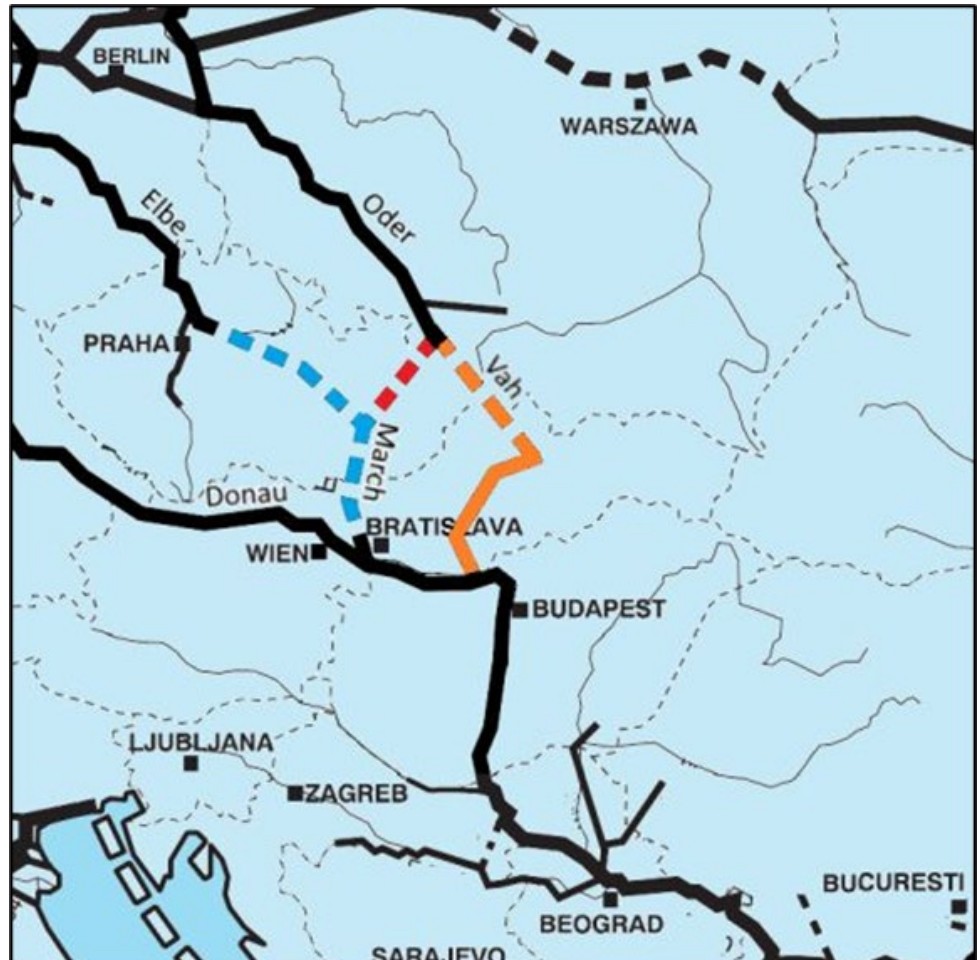

**Figure 8.** The concept of the Danube–Oder–Elbe Canal, including the Váh River, resulting in a substantial NE extension of the Rhine–Main–Danube corridor. Source: https://meta.eeb.org/2021/02/02/controversial-plans-for-destructive-danube-oder-elbe-waterway-moving-forward/ accessed on 8 January 2022.

What are the potential dangers of the Danube–Oder–Elbe Canal?

a.   The endangerment and destruction of a total of 400,000 hectares of habitats in 61 protected areas, extreme damage to the aquatic ecosystem;
b.   Irreparable damage to NATURA 2000 sites, destruction of masses of living beings
c.   Violation of international nature conservation conventions;
d.   Due to the reduction of the riverbed level, the drying up of the surrounding areas and, as a result, the emergence of serious drinking water supply problems;
e.   Decreased efficiency of flood protection in some sections due to sewerage, more difficult flood forecasting and predictability;
f.   Increased risk of direct water pollution.

The committee of experts of the Prague Academy of Sciences did not consider it necessary to connect the three rivers from an economic or environmental point of view, but political intentions and short-term economic interests took the upper hand.

From the Polish side, an adjacent project has been proposed, the closure of another missing link through the construction of the 93 kilometres-long Vistula–Oder Canal to strengthen inland waterways, helping to boost water traffic between the industrial cities of Silesia and the ports of Szczecin and Świnoujście in West Pomeranian Voivodeship. According to initial expert estimates, the construction of the Silesian waterway could cost PLN 11 billion (2.4 billion Euro). Although the project's environmental impact assessment is not yet available, it is likely that the connection would have more negative effects on

the environmental ecosystems than benefits, and the dangers of the ongoing mining activities should be taken into consideration.

### 6.2. Waterway Corridors Are Corridors for Invasive Species—A Risk Enhanced by Climate Change

Inland waterway corridors increase the potential for aquatic species to conquer new habitats in Europe, which can be further enhanced by the construction of new canals to close the missing links, as we illustrated in the abovementioned cases. There is an already existing complex network of inland waterways connecting previously isolated hydrographic catchment areas, which act as invasion corridors for non-native species, many of which may gain ecological advantage through their greater adaptability to new environmental conditions such as warmer water temperatures, oxygen deficiency, water level changes, reduced biological filter capacity, etc. [21]. Leuven et.al. [22] studied the dispersal of macroinvertebrate invasive species in the river Rhine and concluded that the recent Rhine–Main–Danube waterway (southern invasion corridor) can be regarded as the main gateway for the dispersal of invasive, non-indigenous species, some of which dominated the habitats of the littoral zone, seriously harming the biodiversity and the ecological integrity of the river by displacing less resilient native species.

### 6.3. The Case of the Lake Balaton–Sió Canal–Danube Waterway

Lake Balaton is the largest lake in Hungary and Central Europe, the largest feeder of which is the Zala River. Since the lake did not have a permanent, natural outflow, its water level fluctuated greatly over time; its surface has been significantly reduced compared to the original due to the filling work of the inflowing watercourses. The water level is now regulated by the Sió lock, which drains part of the water flow through the Sió canal to the Danube (Figure 9). On the site of today's Siófok, the largest town on the south shore, before the 19th century, there was a swampy, reedy area, with a water level three metres higher than it is now. Although the water from the catchment area of Lake Balaton was drained by natural runoff, it was necessary for the people living in the area to keep the lake open, particularly after the construction of the railway along the southern shoreline of the lake. The final solution was the construction of the Sió riverbed. The first documented regulation took place in 1776. Before it, the Sió Valley was a continuous swamp stretching over three thousand hectares. However, the unregulated small riverbed of Sió was not suitable for transporting even larger amounts of water. In 1858, the construction of the railway line of the Southern Railway Company was started along the southern shore of the lake. The Sió Canal and a sluice were built in 1863, and the water level of the lake decreased by 0.95 m; thus, the natural water flow of Lake Balaton ceased, and it became an artificially regulated lake. The sluice was rebuilt and modernised several times during the last few decades [23,24].

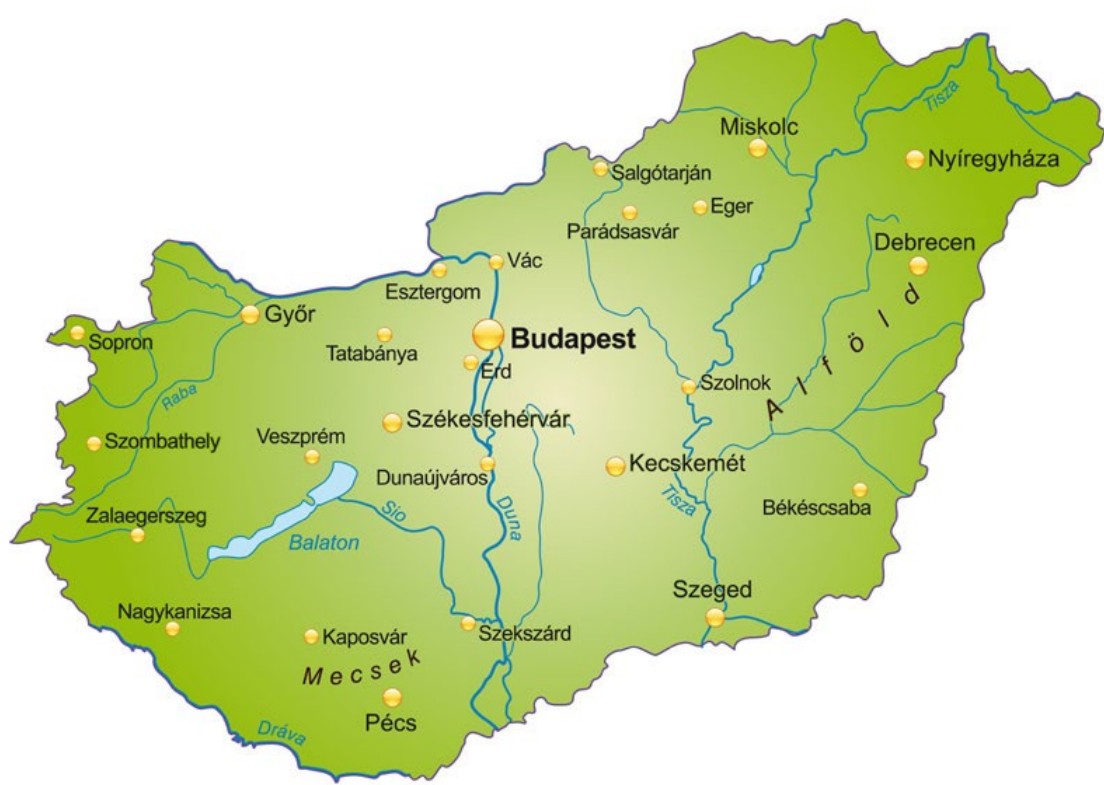

**Figure 9.** Map of Hungary with Lake Balaton and the Sió canal linking the lake to the river Danube. Source: https://magiamiejsc.com/wegry-polozenie-mapa-flaga-stolica-waluta-turystyka.html (accessed on 15 February 2022).

Navigation on the Sió Canal can basically take place in accordance with the Navigation Regulations and the regulations of the waterway operator. In terms of classification, the Sió canal is a quarter-level waterway. Large ships can only sail according to the navigation program announced at the time of the water drainage from Lake Balaton. Only one-way traffic is allowed on the canal at a time; if the ships have passed in that direction, then the opposite direction can start. The average journey time from Siófok to the Danube (in the valley) is 8–12 h and upwards 15–20 h. For longer vessels (over 30 m), a towing device is also required, as this is the only way to manoeuvre safely in bends. In the case of valley convoys, a sliding chain, which is a chain pulled on the bottom of a riverbed, is used at the end of the last vessel to keep the end of the convoy in the middle of the canal, which does not allow the vessel to swing out. During navigation, the rate of water drainage is also changed; in the sections close to Lake Balaton, the higher water drainage results in deeper sailing water, and due to the low bridges in the sections near the Danube, more moderate water drainage is required.

The water flow velocity of the Sió canal is, on average, 1–4 km/h at sea level (faster in the upper section, slower close to the Danube), but in some places, the flow accelerates significantly through the narrowing structures extending into the riverbed (bridges, locks).

## 7. What Are the Possibilities to Replace Fossil Fuels with Electricity or Ecological/Carbon Neutral Fuels in Inland Waterway Transport Compared with Railway?

In maritime transport, great technical advances were accomplished concerning the electric propulsion systems for large cargo vessels. The world's first zero-emission, battery-powered autonomous container ship with a capacity of 120 TEU, Yara Birkeland, was delivered to the Norwegian fertilizer company Yara Norge AS in November 2020



(https://electrek.co/2021/06/08/meet-the-worlds-first-electric-autonomous-container-ship/, accessed on 31 January 2022).

For inland waterways, such fully electric, autonomous cargo vessels have not been developed so far. Inland waterway transport includes both freight and passenger transport, with partly different minimum requirements for flow and riverbed morphology in terms of water depth and manoeuvrability of vessels. Freight transport is classified according to ship classes, such as self-propelled cargo barges, pushed barges, and pushed tankers operated by pusher boats. Vessels used for passenger transport on inland waterways include day trip boats, cabin vessels, and small watercrafts/sporting boats. Although the use of ecological fuels and electric propulsion is growing, this trend is more pronounced for smaller vessels than large cargo barges. However, alternative fuels and propulsion systems such as biogas, methanol-to-gasoline, biodiesel, hydrogen, and battery-electric propulsion (for smaller vessels) can be regarded as promising future options [25]. Instead of pure electric propulsion systems for large cargo barges on rivers hybrid diesel (Diesel-Battery-Electric-Propeller) can be recommended [26]. It is important to emphasize the connection between the volume of transport and the $CO_2$ emission/tkm on inland waterways: the higher the TEU, the lower the $CO_2$ emission/tkm.

As far as electric rail transport is concerned, the EU-27 average was 56% in 2019, while Switzerland was the only European country where 100% of railways were electrified. Among the European Union member states, Luxembourg had 91% electrified railway systems. The usable railway lines in the EU-27 had a combined length of 200,161 kilometres in 2019 (Statista Research Department, 2022) (https://www.statista.com/statistics/451522/share-of-the-rail-network-which-is-electrified-in-europe/, accessed on 15 January 2022).

### 8. What Are the Most Important Differences and Similarities between European Inland Waterways and the Waterway Systems in North America in Terms of Environmental, Economic, and Social Benefits and Problems?

The comparison of inland waterways in terms of geographic and climatic conditions, water resources management practices, navigability, environmental problems, hydrological regimes, and socio-economic conditions is particularly justified in view of climate change. Climate change is expected to bring about significant changes in the field of surface water management since it also changes hydrological conditions [27,28]. Most models anticipate a significant increase in annual mean temperature, which is expected to be greatest during the summer season. Based on the results, the periodic distribution of the annual precipitation is expected to increase in winter and decrease in summer [28,29]. As a result of climate change, extreme weather events may become more frequent, causing unprecedented rainfall or extreme droughts over time. The reduction of snowpack and glaciers resulting in low summer flows can be expected, with an adverse impact on the navigability of watercourses, aquatic ecosystems, agriculture, and drinking water supply, while areas with substantial groundwater resources will be less affected [6,27]. Considering the differences among inland waterways in other geographic locations and climate zones, we make a very short comparison regarding the benefits and problems between the European and the most important inland waterway systems of North America. These waterways are situated in different continents and climate zones and often connect several areas with variable demographic, social, economic, technological, and environmental conditions.

The most important networks of inland waterways in North America include the connection of the Great Lakes (Lake Superior, Lake Michigan, Lake Huron, Lake Erie, Lake Ontario) and the Atlantic Ocean through the St. Lawrence Waterway and the vast Mississippi River System (also referred to as the Western Rivers), which drains 59% of all rivers in the United States within a drainage basin of 3,224,535 square kilometres (Figure 10) (https://www.epa.gov/ms-htf/mississippiatchafalaya-river-basin-marb, accessed on 19 May 2022). The two waterways have been connected since the 19th century, first

through the Illinois and Michigan Canal (I&M) opened in 1848, and then in 1900 through the Chicago Sanitary and Ship Canal and later via the Illinois Waterway system (https://www.mvr.usace.army.mil/Missions/Recreation/Illinois-Waterway/, accessed on 19 May 2022) stretching from the mouth of the Calumet River at Chicago to the mouth of the Illinois River at Grafton, providing navigable waterways of 541 km [30]. Due to the drop of the Illinois Waterway from 176 m above sea level at Lake Michigan to 128 m at the Mississippi River at Grafton, a system of eight locks and dams had to be constructed to provide the lift for traffic along the waterway. The system is managed by the Army Corps of Engineers to regulate the water flow from Lake Michigan to the Mississippi River System.

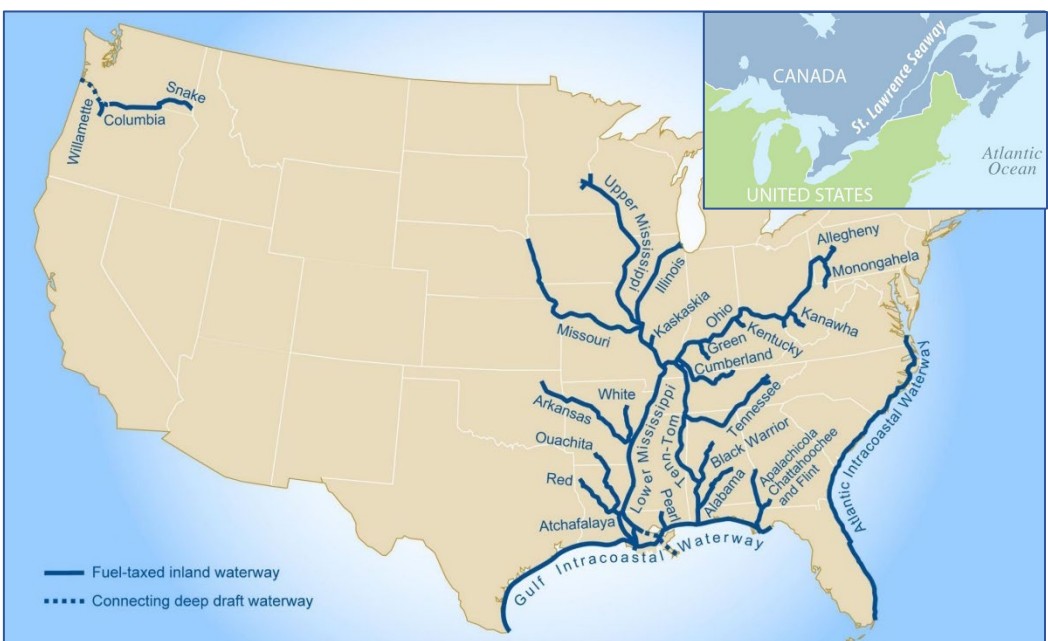

**Figure 10.** The inland waterways of the United States, including the St. Laurence Seaway, shared between the United States and Canada. Source: U.S. Army Corps of Engineers and GAO (https://grains.org/grains-go-with-the-flow-u-s-inland-waterway-system-operating-normally-during-covid-19/, accessed on 20 May 2022).

The well-developed, huge network of inland waterways connected to natural ports throughout coastal locations makes water transport the most economical way to deliver large amounts of bulk commodities such as fuel, timber, ore, building material, agricultural products (grain) to large distances in a sustainable way with low GHG emissions. The more than 19,000 kilometres-long system of navigable routes is maintained by the US Army Corps of Engineers, connecting a vast area of expansive farmlands, and has contributed greatly to the development of agriculture and settlements. The waterway system is truly multifunctional; beyond freight transport, there are other benefits, which include the use of surface waters for irrigation, drinking water supply, personal transport, environmental remediation such as the Wetland Reserve Program (WRP) (https://www.federalregister.gov/documents/1995/06/01/95-13161/wetlands-reserve-program, accessed on 20 May 2022), creating/restoring habitats, tourism, fishing, etc., and it provides excellent opportunities for new energy- and water-saving food production technologies, such as aquaculture and aquaponics [31,32].

However, the infrastructure of this system requires maintenance, such as the dredging of ports and rivers and the control and maintenance of dams, levees, and locks. Although compared to European waterways, there are no missing links between the different parts of the US waterway system, due to the age of the infrastructure, there are several technical problems that are particularly pronounced on the Mississippi and Ohio rivers

and the Illinois waterways with expansive lock systems. Since most of these locks were constructed in the early 20th century and have an expected lifetime of 50 years, they are still in operation. Many of them are over 80 years old, causing an increase in mechanical breakdowns for more than a decade. There is an estimated bill of roughly 13 billion dollars to improve the inland waterway infrastructure in the United States [33–35], (https://worldview.stratfor.com/article/united-states-problem-aging-infrastructure-inland-waterways, accessed on 13 May 2022). Sustainable waterway renewal often requires changes in capacity (expansion or reduction), where transportation aims are combined with ecosystem conservation and a holistic approach to regional development [17].

The other issue, which might become a more serious problem in the future due to climate change, is the amount of water being released into the Illinois River, creating conflicts between lake and river interests. When Lake Michigan water levels are high, a larger amount of water needs to be released into the Illinois River, while at low water levels, the key stakeholders of Lake Michigan want to restrict the flow. Therefore, an international treaty regulates the flow, since Canada also has an interest in the management of Lake Michigan, which is connected to the lakes Huron, Erie, and Ontario [36].

There are several environmental problems arising in connection with the establishment and the operation of inland waterways. The sustainable establishment, expansion, and operation of inland waterways must protect the ecological functions of river systems in terms of channel continuity, riparian and floodplain connectivity, flow regime, and biodiversity. Therefore, a certain level of trade-off may be required between the intensity of waterway exploitation, infrastructure maintenance and renewal, and nature conservation, taking into consideration the possible future climatic and hydrological uncertainties [37].

One serious environmental problem is the salinization and alkalinization of watercourses and lakes. A new, so far unique study, carried out by a research group at the University of Maryland, assessed long-term changes in freshwater salinity and pH at the continental scale based on data recorded at 232 U.S. Geological Survey monitoring sites and found significant increases in both properties, which they attributed to road de-icers, irrigation runoff, sewage, potash, soil cation exchange, mining, and the presence of easily weathered minerals used in agriculture (lime) and building materials [38,39]. The researchers coined the concept of Freshwater Salinization Syndrome, which showed a continuous increase in specific conductance, pH, alkalinity, and base cations, endangering ecosystem services such as safe drinking water, contaminant retention, and biodiversity.

Debris and plastic waste continuously enter the Mississippi River and its tributaries, threatening the health of ecosystems. A significant portion of the approximately 11 million metric tons of plastic waste that enter the oceans every year is transported by rivers. According to a report by UNEP, 75% of retrieved items were plastic, including cigarette butts, food wrappers, and beverage bottles [40–42].

Invasive animal and plant species in the Great Lakes is another issue that requires control, keeping in mind that these invaders may replace indigenous populations and that, once established, it is extremely difficult to control their spread. The invasive species have continuously changed the lake ecosystems during the past two centuries. At least 25 invasive animal species have entered the Great Lakes since the 1800s, and the most prominent of these are the following: *Brachionus leydigii,* a tiny, non-native invertebrate rotifer and a type of zooplankton; *Thermocyclops crassus*, a thermophilic cyclopoid with a preference for eutrophic waters; Asian carp (*Cyprinus carpio*), which is native to Europe and Asia; round goby (*Neogobius melanostomus*) a euryhaline bottom-dwelling fish native to central Eurasia; sea lamprey (*Petromyzon marinus*), a parasitic lamprey sometimes referred to as the "vampire fish"; Eurasian ruffe (*Gymnocephalus cernua*), which is native to Eurasia; alewife (*Alosa pseudoharengus*), an anadromous species of herring; zebra mussel (*Dreissena polymorpha*), a small freshwater mussel that is native to the lakes of southern Russia and Ukraine; and spiny water flea (*Bythotrephes longimanus*), a small crustacean and zooplankton native to Great Britain, with its home range extending through Northern Europe and east to the Caspian Sea. The ecosystems of the Great Lakes are also threatened by fast-

growing invasive plants, which displace the native plants that support wildlife habitat and prevent erosion. The most prominent species are the folling: common reed (*Phragmites communis*), reed canary grass (*Phalaris arundinacea*), purple loosestrife (*Lythrum salicaria* L.), curly pondweed (*Potamogeton crispus*), Eurasian milfoil (*Myriophyllum spicatum*), frog-bit (*Hydrocharis morsus-ranae*), and non-native cattail.

Thirty percent of invasive species in the Great Lakes have been introduced through ship ballast water, while others migrated through the waterways connected to the Great Lakes. Many stakeholders of the Great Lakes demanded the permanent closure of the canals (the extensive Chicago Waterway System) connecting to the Illinois River to stop the invasion of Asian carp. https://www.epa.gov/greatlakes/invasive-species-great-lakes accessed on the 15 May 2022.

Comparing the North American and the European Waterway systems, the most significant difference is that in Europe, many connections that could connect the waterways are missing, while in the United States, there is a well-developed waterway network. Even if this network needs an extensive renewal, it is a well-functioning waterway, suitable for transportation of vast amounts of bulk commodities, serving the surrounding agricultural enterprises, providing personal transport, recreation facilities, and many other functions for the benefit of the inhabitants. In Europe, the largest waterway is the Rhine–Main–Danube corridor, where the Danube is not fully navigable during the whole year since low water levels or ice prevent or restrict navigation [43]. Considering the environmental problems in terms of pollution, inadequate land use, the loss of ecosystem services, and the risk of invasive species, the key issues are very similar, but in the case of the waterways of the United States, it is a great advantage that the problems can be solved in the same country, while in Europe, international negotiations are needed to find a common solution for conflicting interests. Regarding the impacts of climate change, it is causing serious problems for the waterway systems in both Europe and the United States due to extreme weather, but during the last decades, some extreme events, such as tornadoes, floods, and wildfires, occurred more frequently in the USA.

## 9. Is There an Ecologically Sustainable Solution for a Pan-European Inland Waterway Network?

The advantages of inland waterways cannot always be exploited, as inland waterway transport services are only available where the right quality of route is available. The waterway network is far from being as dense as the public road and rail network, and this, of course, has a bearing on transport volumes.

Climate change is continuing, and new record levels in terms of temperature, global sea level change, droughts, and the decline of polar ice sheets have been recorded, which caused vast damages in ecosystem services and triggered the investigation of measures to mitigate the adverse impact of climate change. These events are threatening the existing ecosystems, which must be protected, and all future development must take into consideration the resilience of these ecosystems.

The environmental impact of shipping should be considered in conjunction with the effects of the development of infrastructure for inland waterway transport, including the construction of ports and access to ports and the emission of ships. Further burdening the ecosystem of rivers with adverse environmental impacts caused by high-cost interventions that are important for their continued maintenance cannot be supported (Wang et al. 2020). When examining the alternatives, the connection possibilities of the waterway and the railway must be covered, and efforts must be made to establish and develop connections between the various modes of transport (inland waterways, railways, roads, i.e., inter- and multimodality) and to take into consideration any necessary sections with solutions to replace capacity (e.g., transporting goods by rail during low water periods). The analysis of development alternatives must pay attention to the extent to which the ideas represent sustainable regional interests. The abovementioned ELOHA model needs to be completed with thorough environmental risk assessment, environmental consequence

analysis, and assessment of ecosystems and ecosystem services and their remediation strategies in case of a flow alteration of watercourses serving as inland waterways (Figure 11).

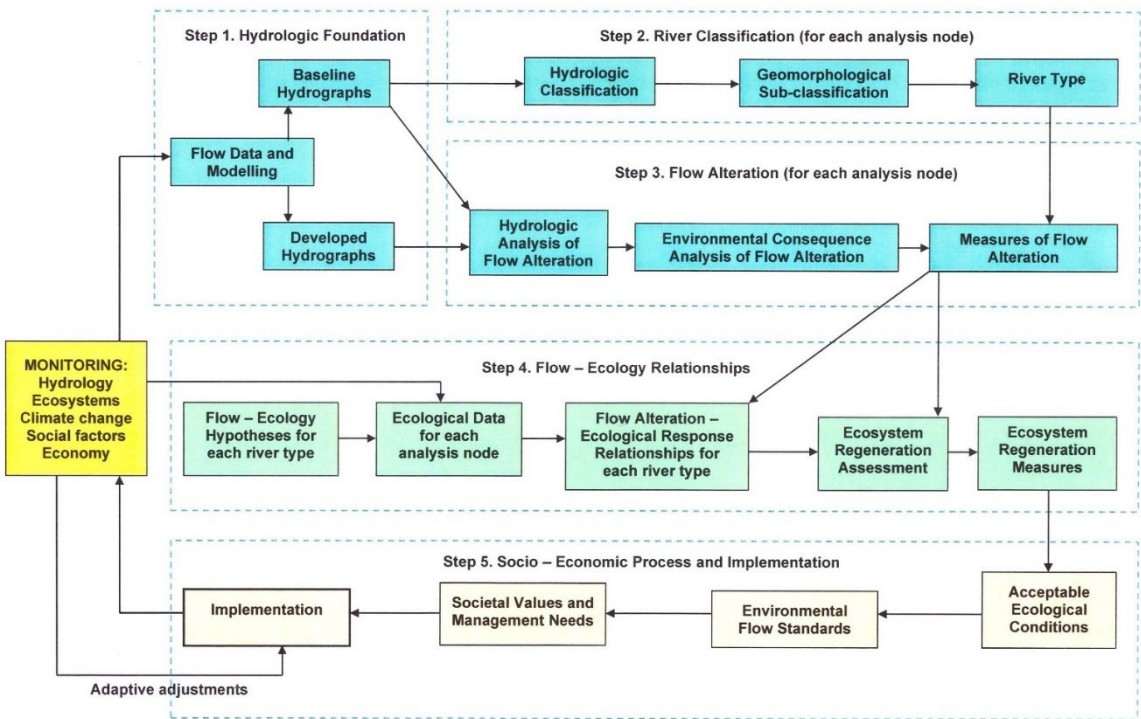

**Figure 11.** The concept of sustainable construction of inland waterways. The complex ELOHA method can be helpful in the construction of ecologically viable improvements of inland waterways but must be completed with the ecosystem approach Source: Modified and redrawn after [11].

In order to reduce pollution, we recommend prioritizing organic farming along the inland waterways, as well as traditional farming methods (e.g., floodplain farming, degree farming) over intensive agriculture. Due to the extreme weather conditions caused by climate change, high-intensity precipitation and flood-like rains are becoming more frequent, with which a higher proportion of plant protection and nutrient replenishment residues are washed into living waters.

As part of the transport corridor network, the provision of a waterway that meets European requirements can be supported via environmentally and nature-friendly solutions based on a detailed economic and energy efficiency study that presupposes the fair and proportionate use of waterway goods and related ecosystem services.

## 10. Conclusions

Considering the substantial differences in the GHG efficiency of motorised transport modes in Europe, the need of shifting transport to the most efficient modes became obvious and resulted in new transport strategies in the European Union, targeting the relatively low carbon alternatives of rail and waterborne transport for both passengers and freight.

While rail transport and aviation have greatly improved their GHG efficiency during the last decades, only small improvements or sometimes stagnation could be observed in other transport modes. For rail, the improvements are a result of the electrification of the rail network and the continuously declining carbon intensity of the EU's electricity mix. The shift from one transport mode to another is strongly limited by distance, geomorphology, infrastructure, and available time.

There are some contradictions regarding the GHG efficiency of rail and waterborne transport, which claim that the waterways are more environment friendly in terms of emissions, while other studies support the rail transport as the most GHG efficient form of transport. This is dependent on the extent of fossil fuel use, the source of electricity (fossil or renewable), and the volume of transported goods. Furthermore, rail and waterway transport modes can be used only between transport hubs, such as ports and freight terminals, and therefore, are viable only in combination with other modes.

Although the EU's modal shift policy is confirmed by these data, not all modes are equally suited to all transport tasks, which limits the possibility to substitute one mode of transport for another due to geography, availability of infrastructure, and time criticality.

The creation of a complete pan-European inland waterway network requires the closure of the so-called missing links. However, this might cause serious damage to both aquatic and terrestrial ecosystems in terms of lost habitats, increased flow and erosion, damaged ecosystems, and lost ecosystem services. However, a well-planned extension of the inland waterway network according to the modified ELOHA method and the establishment of a multimodal transport system can offer a viable compromise for a well-developed inland waterway network in Europe.

**Author Contributions:** Conceptualization and writing original draft preparation, S.A.N. and A.T.; methodology, formal analysis, and investigation, L.B., B.L. and L.S.; writing—review and editing, L.B., B.L. and L.S. All authors have read and agreed to the published version of the manuscript.

**Funding:** This research received no external funding.

**Conflicts of Interest:** The authors declare no conflicts of interest.

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
