# Peer review of "Environmental Viability Analysis of Connected European Inland–Marine Waterways and Their Services in View of Climate Change"

_atmosphere, doi:10.3390/atmos13060951_

Round 1

Reviewer 1 Report

Review of Atmosphere:1623175
Title: Environmental Viability Analysis of Connected European Inland – 
Marine Waterways and Their Services in View of Climate Change
by Némethy et al.

This is not a scientific paper.  It presents no equations, no new measurements, no quantitative analysis, and in short, offers nothing original.   Instead, it presents a laundry list of environmental and economic banalities, as it bandies about platitudes regarding climate change, air pollution, heavy precipitation, droughts, melting glaciers, floods, variable flow, aquatic ecosystems, wetlands, invasion corridors, water quality and countless more- even “willow, poplar, alder and ash populations”.  The paper presents no comparison of the European Inland Waterway with analogous systems elsewhere, such as the huge Inland Waterway System of the USA. Nobody needs this.

Illustrations are poor and none are unoriginal.  Writing is too small on Fig. 1; the redundant inset on Fig. 2 is unnecessary, confusing and not even mentioned; Figures 1, 4 and 6 lack scale bars, no figure highlights the location of other figures, many important place names mentioned in the paper are not highlighted on any figure, etc. 

Recommendation: Reject.  Writing is clear and a magazine for general readers might consider it.

Author Response

Dear Reviewer,

Thank you very much for your valuable review. We have rewritten the article and completed the text and the figures according to your critical remarks.

Reviewer 2 Report

The paper described Environmental Viability Analysis of Connected European Inland – Marine Waterways and Their Services in View of Climate Change was given in this paper. This study was analyzed the Danube – Tisza – Rhine – Black Sea inland waterway corridors and their branches (including the future possibilities of Lake Balaton – Danube waterway), and the inland waterways of Poland (mainly the Oder, the Vistula, and the Vistula-Oder waterway). Landscape conservation, green infrastructure development, water replenishment, and ecosystem reconstruction were all offered as model structures.

The manuscript is well formulated. The results are presented clearly and they are well discussed. The manuscript is meaningful, organized and therefore I recommend it for publication in the Atmosphere.

Author Response

Dear Reviewer,

Thank you very much for your valuable review.

Reviewer 3 Report

Review: Environmental Viability Analysis of Connected European In- and - Marine Waterways and Their Services in View of Climate Change.

The work discusses how inland waterways and their connections to marine transport systems constitute numerous inherent environmental hazards. The authors attempted to show the environmental viability of these waterways based on the right situational assessment and precise deployment of ecosystem services.  

Overall, the writing and quality of communication are well articulated. It clearly expresses its information and, the authors paid attention to readability and proper use of language. 

However, I struggle with the novelty of this work. The authors did not build a strong justification for their research. Your objective and premises were loosely built as the rest of the work. 

Please look through your work carefully paying attention to the following: 

  • The abstract is too long; perhaps the authors could shorten it and make it more concise i.e. a brief introduction, the purpose, the problem, methodology, and result. 
  • The introduction/theoretical background is inadequate. Necessary parameters to set the tone for the work were missing i.e. motivation or justification for work, aim or objective of work and the single research question brought up is too broad and does not give adequate justification for the research.
  • Line 41-43. Although this information is generally known in the maritime industry, it still needs to be referenced.
  • The referencing style is wrong and does not fit the requirement of the journal. Citations and referencing are grossly missing from the introduction and other parts of the work, especially the introduction. Please cite your work appropriately. 
  • The methodology section was clearly missing. Please write how you were able to collect your data and how they were analyzed. Even though it is a review, certain academic guidelines have to be followed. 
  • You need to present your findings and results clearly. Analyzed and interpret the results appropriately. It is critical to bring out the contribution of the work to the body of knowledge by comparing your work to others out there (there should be a few with a similar premise). Please do this by expanding your discussions. The conclusion was more of a discussion rather than a conclusion, so, think about it one more time. Re-arrange your work so that the current conclusion and recommendation are part of your discussion. 
  • You might need to create a new conclusion. Look at your work and, what you were set out to do, did you achieve it. You need to state this clearly in your conclusion. Who can use your work and, how will your work help practice and theory? If possible, you can add if there were limitations and how you overcame them and if there are plans for future studies - you can do these in three short paragraphs.

Author Response

Dear Reviewer,

Thank you very much for your valuable review. We have restructured, corrected  and completed our article according to your suggestions. 

Yours sincerely: 

Sándor Némethy

Round 2

Reviewer 1 Report

Authors have eliminated much of the "fluff" in their original version, but have done a poor job describing their response to prior reviewer comments.  Their two sentence "responses"  that include "thanks for your valuable comments" does not suffice.  Moreover, the authors have not (1) provided a master figure showing the location of their other maps, nor 2) addressed the benefits and problems with other inland waterway systems, such as that in the USA. So, this is not a scientific study, and because of item 2, it is hardly a "review" of what is known.

Author Response

Dear Reviewer,

Thank you for your suggestions, we tried to improve the article by completing it with a master figure showing the location of the other maps and made a (fairly short)comparison of the benefits and problems of European waterway with the Waterway system of the USA, which, indeed is the most relevant regarding its structure, size, geography and environmental issues. We considered to make more comparisions with other waterways such as the waterway systems of the Amazon-river, the Nile in Africa, or the Jangtze-river in China - but this would have been too long for this study... 

Reviewer 3 Report

I am satisfied with the changes made by the authors

Author Response

Dear Reviewer,

Thank you for your suggestions concerning changes in English - we have gone through the article and made the necessary connections, even for the recently added paragraphs of the study.